# Generation and characterization of monoclonal antibodies against pathologically phosphorylated TDP-43

Paula Castellanos Otero[1], Tiffany W. Todd[1], Wei Shao[1], Caroline J. Jones[1], Kexin Huang[1], Lillian M. Daughrity[1], Mei Yue[1], Udit Sheth[1,2], Tania F. Gendron[1,2], Mercedes Prudencio[1,2], Björn Oskarsson[3], Dennis W. Dickson[1,2], Leonard Petrucelli[1,2]*, Yong-Jie Zhang[1,2]*

1 Department of Neuroscience, Mayo Clinic, Jacksonville, Florida, United States of America, 2 Neurobiology of Disease Graduate Program, Mayo Graduate School, Mayo Clinic College of Medicine, Rochester, Minnesota, United States of America, 3 Department of Neurology, Mayo Clinic, Jacksonville, Florida, United States of America

* petrucelli.leonard@mayo.edu (LP); zhang.yongjie@mayo.edu (Y-JZ)

**Data Availability Statement:** All relevant data are within the manuscript and its Supporting Information files.

## Abstract

Inclusions containing TAR DNA binding protein 43 (TDP-43) are a pathological hallmark of frontotemporal dementia (FTD) and amyotrophic lateral sclerosis (ALS). One of the disease-specific features of TDP-43 inclusions is the aberrant phosphorylation of TDP-43 at serines 409/410 (pS409/410). Here, we developed rabbit monoclonal antibodies (mAbs) that specifically detect pS409/410-TDP-43 in multiple model systems and FTD/ALS patient samples. Specifically, we identified three mAbs (26H10, 2E9 and 23A1) from spleen B cell clones that exhibit high specificity and sensitivity to pS409/410-TDP-43 peptides in an ELISA assay. Biochemical analyses revealed that pS409/410 of recombinant TDP-43 and of exogenous 25 kDa TDP-43 C-terminal fragments in cultured HEK293T cells are detected by all three mAbs. Moreover, the mAbs detect pS409/410-positive TDP-43 inclusions in the brains of FTD/ALS patients and mouse models of TDP-43 proteinopathy by immunohistochemistry. Our findings indicate that these mAbs are a valuable resource for investigating TDP-43 pathology both *in vitro* and *in vivo*.

## Introduction

Inclusions containing TAR-DNA binding protein of 43 kDa (TDP-43) are a hallmark of multiple neurodegenerative diseases collectively referred to as TDP-43 proteinopathies. These misfolded aggregates have been observed in 45% of frontotemporal dementia (FTD) cases and 97% of amyotrophic lateral sclerosis (ALS) cases [1–3]. TDP-43 is a highly conserved DNA/RNA binding protein composed of a N-terminal domain, two highly conserved tandem RNA recognition motifs, and a prion-like C-terminal domain [3,4]. As a heterogeneous nuclear ribonucleoprotein (hnRNP), TDP-43 regulates RNA metabolism including transcription, splicing, and microRNA biogenesis and processing [5]. Under normal conditions, TDP-43

**Funding:** This work was supported by the National Institutes of Health [R35NS097273 (L.P.); P01NS084974 (D.W.D., T.F.G., L.P., Y.-J.Z.); P01NS099114 (T.F.G., L.P.); U54NS123743 (L.P.); RF1AG062077 (L.P.); RF1AG062171 (L.P.); R01NS117461 (T.F.G., Y.-J.Z), 1R21NS127331 (Y.-J.Z)], the Mayo Clinic Foundation (L.P.), the Robert Packard Center for ALS Research at Johns Hopkins (L.P.), and the Target ALS Foundation (L. P., Y.-J.Z.). The funders had no role in study design, data collection and analysis, decision to publish, or preparation of the manuscript.

**Competing interests:** The authors declare no competing interests.

predominantly localizes to the nucleus of neurons and glia in the central nervous system. Under disease conditions, however, TDP-43 forms insoluble aggregates in the cytoplasm, thus leading to a substantial nuclear depletion of TDP-43 [1,2,6]. Notably, pathological TDP-43 exhibits post-translational modifications such as hyperphosphorylation, ubiquitination, sumoylation, acetylation and fragmentation [7,8]. For example, phosphorylation of TDP-43 at serines 409/410 (pS409/410) is a pathological hallmark of TDP-43 proteinopathies [9]. It has been reported that this modification impedes TDP-43 proteasomal degradation, accelerates TDP-43 inclusion formation, and affects TDP-43 localization and function [10–12]. The morpholgy and distribution of phosphorylated TDP-43 inclusions are used for subclassifying frontotemporal lobar degeneration with TDP-43-positive inclusions (FTLD-TDP) cases into five recognized subtypes [13,14].

Due to the importance of TDP-43 S409/410 phosphorylation in disease pathogenesis and pathology, tools to measure pS409/410-TDP-43 using various methods are crucial. Thus, although a few pS409/410-TDP-43 antibodies already exist [7,9,15], we generated and characterized three new rabbit monoclonal antibodies (26H10, 2E9 and 23A1) against pS409/410-TDP-43. These mAbs recognize pS409/410-TDP-43 with high sensitivity and specificity, as evidenced by assays using *in vitro* recombinant proteins, cultured cells, TDP-43 mouse models, and FTD/ALS patient tissue. We have thus generated valuable new resources that can be utilized in a variety of research and diagnostic applications and hold important therapeutic potential.

## Materials and methods

### Generation of antibodies

All procedures for antibody generation were done by Labcorp Laboratories and conducted in compliance with the U.S. Department of Agriculture's (USDA) Animal Welfare Act (9 CFR Parts 1, 2, and 3); the Guide for the Care and Use of Laboratory Animals (Institute of Laboratory Animal Resources, National Academy Press, Washington, D.C., 2011); and the National Institutes of Health, Office of Laboratory Animal Welfare. Whenever possible, procedures in this study were designed to avoid or minimize discomfort, distress, and pain to animals. In brief, to generate rabbit monoclonal antibodies against pS409/410-TDP-43, a rabbit was immunized with keyhole limpet hemocyanin-conjugated pS409/410-TDP-43 peptides (Ac-CSMDSK[pS][pS]GWGM-OH). Blood was collected prior to immunization (pre-bleed) and after immunization (bleed). After confirming that the bleed exhibited specificity and sensitivity to pS409/410-TDP-43, the rabbit was euthanized by injection with ketamine/xylazine followed by $CO_2$ inhalation and exsanguination. The spleen was harvested and the splenocytes were processed for B cell sorting by fluorescent activated cell sorting. Culture supernatants were used for examining the specificity and sensitivity of each B cell clone against pS409/410-TDP-43. The heavy and light chain genes from three B cell clones (26H10, 2E9 and 23A1) were cloned into pcDNA3.4 and pkcDNA plasmids. To produce purified monoclonal antibodies, the light and heavy chain plasmids were co-transfected into Expi293F cells using PEI transfection reagent according to the manufacturers' instructions. Four days after transfection, the culture media was collected for antibody purification using Repligen CaptivA HF resin.

### *In vitro* phosphorylation of recombinant TDP-43

The *in vitro* phosphorylation of recombinant TDP-43 (rTDP-43) proteins was performed as previously reported [16]. In brief, 0.1 μg/μL rTDP-43 was incubated with or without 1 μL CK1 (P6030S, New England Biolabs) in 1× reaction buffer (20 mM Tris, 50 mM KCl, 10 mM

MgCl$_2$, 20 mM ATP, pH 7.5). The total volume of the reaction was 100 μL. The reactions were incubated at 30˚C for 14 h.

## Dot blots

Samples were prepared with Tris-Glycine SDS buffer (Novex, Life Technologies) containing 10% β-ME at a 1:1 ratio (v/v) and heated at 95˚C for 5 min. Five microliters of the sample was loaded to a 0.45 μm nitrocellulose blotting membrane (Cytiva-Amerstam™ Protan™) and incubated at 37˚C for 30 min. Membranes were blocked with 5% non-fat dry milk in Tris-buffered saline (TBS) plus 0.1% Tween 20 (TBST) for 1 h and then incubated with B cell supernatant (1:100) or mAbs (1:500) overnight at 4˚C. Membranes were then washed in TBST three times for 10 min each and incubated for 1 h with donkey anti-rabbit IgG antibody conjugated to horseradish peroxidase (1:5000; Jackson ImmunoResearch) for 1 h at room temperature. Protein expression was visualized by enhanced chemiluminescence treatment using Western Lightning Plus-ECL (Perkin Elmer).

## Generation of plasmids

To generate plasmids of GFP-TDP-43$_{1-408}$ and GFP-TDP-43$_{1-408\text{-NLSmut}}$, the relevant 408-amino acid DNA fragments were amplified from previously described plasmids containing TDP-43 and TDP-43$_{\text{NLSmut}}$ [17]. These TDP-43 fragments were then cloned into pEGFP-C1 (Clontech Laboratories) using BamH1/XbaI restriction sites. The sequences of the plasmids were verified by sequence analysis.

## Cell culture and transfection

Human embryonic kidney 293T (HEK293T) cells were grown in Opti-Mem plus 10% fetal bovine serum and 1% penicillin–streptomycin. For transfection, HEK293T cells were seeded at a density of $1 \times 10^6$ cells/well in 6-well plates. After 24 h, cells were transfected with 2 μg of either GFP, GFP-TDP-25 or GFP-TDP-25-S409A/S410A (GFP-TDP-25$_{\text{mut}}$) or GFP-TDP-43, GFP-TDP-43$_{1-408}$, GFP-TDP-43$_{\text{NLSmut}}$, or GFP-TDP-43$_{1-408\text{-NLSmut}}$ using Lipofectamine 2000 (Thermo Fisher Scientific, 11668500) according to the manufacturers' instructions. Cells were harvested for immunoblot analysis 48 h post-transfection.

## Preparation of cell lysates

Cell pellets were lysed in co-IP buffer (50 mM Tris–HCl, pH 7.4, 300 mM NaCl, 1% Triton X-100, 5 mM EDTA) plus 2% SDS and both protease and phosphatase inhibitors, sonicated on ice, and then centrifuged at $16,000 \times g$ for 20 min. Supernatants were saved as cell lysates. The protein concentration of lysates was determined by BCA assay (Thermo Scientific), and samples were then subjected to immunoblot analysis.

## Immunoblotting

Samples were prepared with Tris-glycine SDS buffer (Novex, Life Technologies) containing 10% β-ME at a 1:1 ratio (v/v) and heated at 95˚C for 5 min. Equal amounts of protein were loaded into 10-well 4–20% Tris-glycine gels (Novex). After transfer, blots were blocked with 5% nonfat dry milk in TBST for 1 h and then incubated with rabbit anti-sera (1:5000), B cell supernatant (1:500), mAbs (26H10, 2E9 and 23A1; 1:500), rabbit polyclonal GFP antibody (1:1000, Thermo Fisher Scientific, A-11122), rabbit polyclonal TDP-43 antibody (1:1000, Proteintech, 12892-1-AP) or mouse monoclonal GAPDH antibody (1:5000, Meridian Bioscience, H86504M) overnight at 4˚C with rocking. Membranes were then washed in TBST three times

for 10 min each and incubated for 1 h with donkey anti-rabbit or anti-mouse IgG antibodies conjugated to horseradish peroxidase (1:5000, Jackson ImmunoResearch) for 1 h at room temperature. Protein expression was visualized by enhanced chemiluminescence treatment using Western Lightning Plus-ECL (Perkin Elmer).

## Animal studies

All procedures using non-transgenic mice (included animals injected with AAV to express normal or expanded *C9orf72* repeats) and rNLS8 mice (which express human TDP-43 that lacks a nuclear localization sequence [hTDP-43ΔNLS] under the control of doxycycline) were performed in accordance with the National Institutes of Health Guide for Care and Use of Experimental Animals and approved by the Mayo Clinic Institutional Animal Care and Use Committee (IACUC) (Protocol numbers A00004784-19 and A00006149-21).

## Tissue processing

Mice were euthanized by injection with ketamine/xylazine, and then a cardiac puncture was performed. Brains were then harvested and hemisected sagittally across the midline. Sagittal hemibrains were immersion fixed in 4% paraformaldehyde, embedded in paraffin, sectioned (5 μm thick), and mounted on glass slides for use in immunofluorescence and immunohistochemistry.

## Human tissues

Post-mortem mid-frontal, motor cortical, and hippocampal tissues from normal controls and patients with FTLD-TDP and ALS were obtained from the Mayo Clinic Florida Brain Bank. Patient information is provided in S1 Table. Written informed consent was obtained before study entry from all subjects or their legal next of kin if they were unable to give written consent, and deidentified biological samples were obtained with Mayo Clinic Institutional Review Board (IRB) approval.

## Immunohistochemistry

Paraffin-embedded brain sections from humans (normal controls, FTLD-TDP subtypes A and B, and ALS cases obtained from Mayo Clinic Jacksonville Brain Bank) and mice (non-transgenic and rNLS8) were deparaffinized in xylene and rehydrated in 100% and 95% ethanol solutions. Antigen retrieval was performed by steaming in 1× sodium citrate buffer (10 mM sodium citrate, 0.05% Tween-20, pH 6.0) for 30 min. Following antigen retrieval, slides were cooled for 15 min and then flushed with distilled water for 15 min. Tissue sections were immunostained with rabbit anti-sera (1:5000), B cell supernatant (1:100), or mAbs (26H10, 2E9 and 23A1, 1:500 unless otherwise indicated) using the DAKO kit (Agilent). Sections were counterstained with hematoxylin, dehydrated through a series of ethanol dilutions and xylene, and coverslipped with Cytoseal mounting media (Thermo Fisher Scientific). Slides were scanned with ScanScope AT2 (Leica Biosystems) at 40× magnification.

## Preparation of urea-soluble fractions

The urea-soluble fraction of tissue sample homogenates was isolated and prepared as we previously described [18]. In brief, frontal cortex tissues were homogenized in cold RIPA buffer (25 mM Tris-HCl [pH 7.6], 150 mM NaCl, 1% sodium deoxycholate, 1% NP-40, 0.1% sodium dodecyl sulfate, and protease and phosphatase inhibitor cocktail), sonicated on ice, and centrifuged at $100,000 \times g$ for 30 min at 4˚C. The supernatant was then collected as the RIPA-soluble

fraction. The pellet was dissolved in urea buffer (30 mM Tris-HCl [pH 8.5], 7 M urea, 2 M thiourea, and 4% CHAPS) for 1 h at room temperature with continuous agitation. Samples were then sonicated and centrifuged at 100,000 × g for 30 min at 22˚C. The resulting supernatant, referred as the urea-soluble or the RIPA-insoluble fraction, was then collected. Protein concentrations of urea-soluble fractions were determined by Bradford assay (ThermoFisher).

## ELISA immunoassay

ELISA immunoassays were performed according to a standard protocol. In brief, ELISA immunoassays were performed in 384-well plates, which were precoated with neutravidin (2 μg/mL) and incubated with biotinylated pS409/410-TDP-43, pS409-TDP-43, pS410-TDP-43, or non-phosphorylated S409/410-TDP-43 peptides (1 μg/mL) overnight at room temperature. B cell supernatants or rabbit mAbs were then added to the wells. After incubation and washing, a secondary goat anti-rabbit IgG antibody conjugated to horseradish peroxidase (1:5000; Jackson ImmunoResearch) was added to the wells for final detection.

## Meso Scale Discovery (MSD) immunoassay

MSD immunoassays to quantify phosphorylated TDP-43 levels in the urea-soluble fraction from frontal cortex was performed as we described previously [19]. In brief, our rabbit mAbs (26H10, 2E9, and 23A1) against pS409/410-TDP-43 (3 μg/mL) were separately used as the capture antibody, and a sulfo-tagged rabbit polyclonal TDP-43 antibody (3 μg/mL, Proteintech, 12892-1-AP) was used as the detection antibody. Urea-soluble fractions were diluted in TBS and then 20 μg of total protein per sample was tested in duplicate wells. MSD QUICKPLEX SQ120 technology was used to acquire the response values corresponding to the intensity of emitted light upon electrochemical stimulation.

## Statistical analysis

Statistical analyses were performed in GraphPad Prism (version 10.0.3.275). The specific tests used are noted in the figure legends as applicable. Immunoassays were performed in duplicate, and all experiments were repeated to ensure reproducibility across trials and samples. No data was excluded from our analyses.

# Results

## Characterization of pS409/410-TDP-43 immunoreactivity in blood from an immunized rabbit

To generate rabbit monoclonal antibodies against pS409/410-TDP-43, we immunized a rabbit with keyhole limpet hemocyanin-conjugated pS409/410-TDP-43 peptides (CSMDSK[pS][pS] GWGM). A pre-bleed and bleed were collected to examine their immunoactivity against pS409/410-TDP-43 (S1 Fig). Immunoblot analysis showed that antibodies in the bleed, but not in the pre-bleed, detect rTDP-43 proteins treated with CK1 (Fig 1A), a kinase known to phosphorylate TDP-43 at S409/410 [20]. In contrast, non-phosphorylated rTDP-43 was not detected, indicating that the bleed specifically recognizes phosphorylated TDP-43. Given that CK1 phosphorylates other serine residues in TDP-43 in addition to S409/410 [16], we next determined whether the bleed specifically detects pS409/410 of TDP-43. Immunoblot analysis of HEK293T cell lysates showed that the bleed detects an ectopic GFP-tagged 25 kDa TDP-43 C-terminal fragment (GFP-TDP-25) (Fig 1B), which is prone to phosphorylation at S409/410 [15]. In contrast, the bleed did not detect a fragment in which serines 409/410 were mutated to alanine (GFP-TDP-25$_{mut}$) to prevent their phosphorylation (Fig 1B). As expected, the pre-

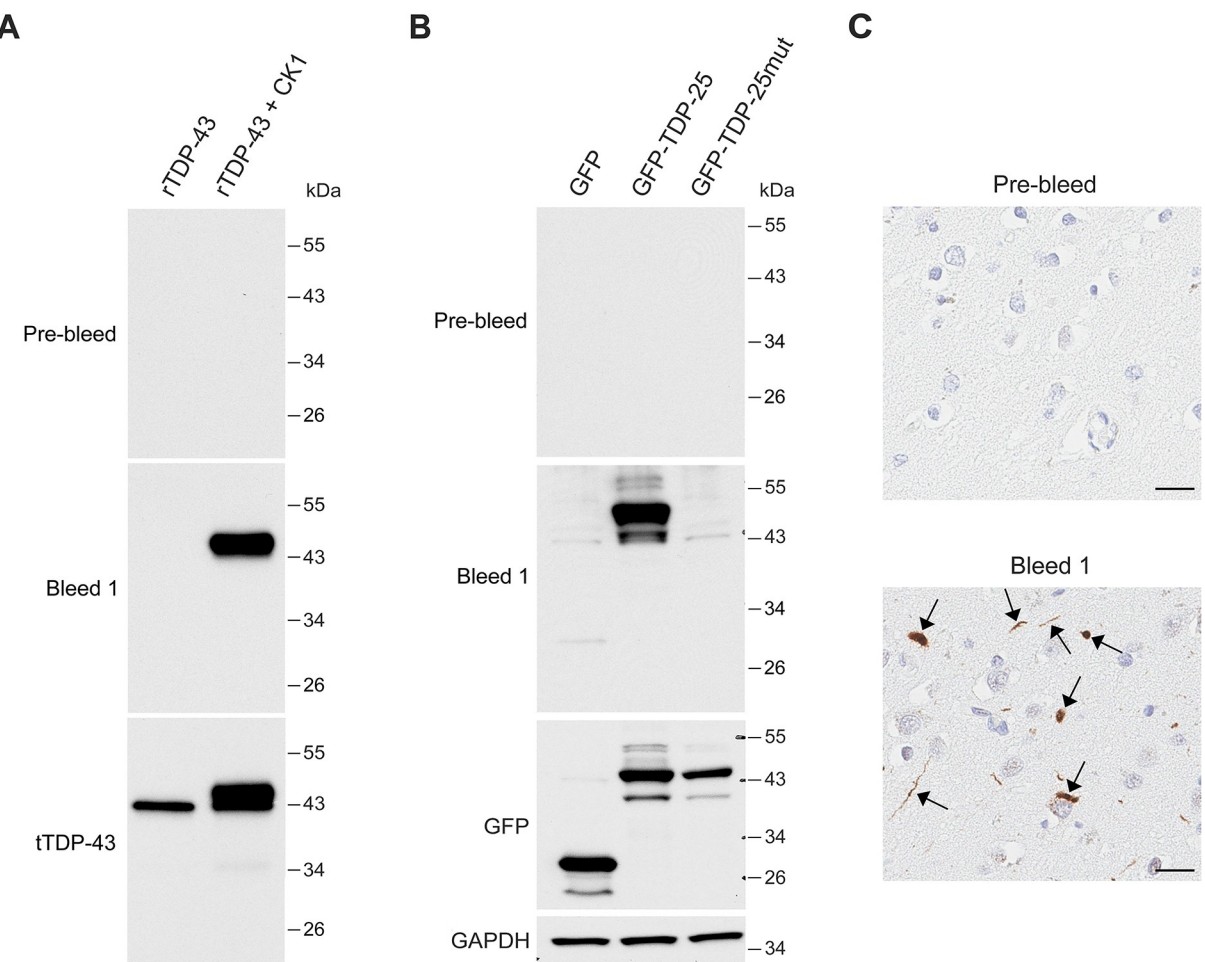

**Fig 1. The rabbit bleed specifically detects pS409/410-TDP-43.** (A) Immunoblot analysis of rTDP43 with or without CK1 treatment using the indicated sera and antibody. (B) Immunoblot analysis of HEK293T lysates expressing GFP, GFP-TDP-25, or GFP-TDP-25$_{mut}$ (S409A/S410A) using the indicated sera and antibodies. GAPDH was used as a loading control. (C) Representative images of immunohistochemical analysis using the indicated sera in the frontal cortex of human FTLD-TDP patients. Arrows mark TDP-43 inclusions. Scale bars are 20 μm.

bleed was unable to detect either GFP-TDP-25 or GFP-TDP-25$_{mut}$ (Fig 1B). We next examined whether the bleed could detect TDP-43 pathology in FTLD-TDP patients. Immunohistochemical analysis showed that the bleed, but not the pre-bleed, detected TDP-43 pathology, including cytoplasmic inclusions and dystrophic neurites, in the frontal cortex of FTLD-TDP patients (Fig 1C). Additionally, no normal nuclear TDP-43 staining was observed with the bleed (Fig 1C). Together, our findings indicate that the rabbit we immunized produced polyclonal antibodies that specifically detect pS409/410-TDP-43 *in vitro*, in cultured cells, and in FTLD-TDP patient tissue.

## Screening B cell clones against pS409/410-TDP-43

To isolate B cell clones and test their immunoactivity against pS409/410-TDP-43, the spleen of the immunized rabbit was harvested and subjected to single B cell sorting using biotinylated pS409/410-TDP-43 peptides as the sorting antigen. We isolated and cultured 2688 B cell clones, and the supernatant from each individual B cell clone was collected to examine its immunoactivity against pS409/410-TDP-43 peptides via ELISA immunoassay (S1 Fig).

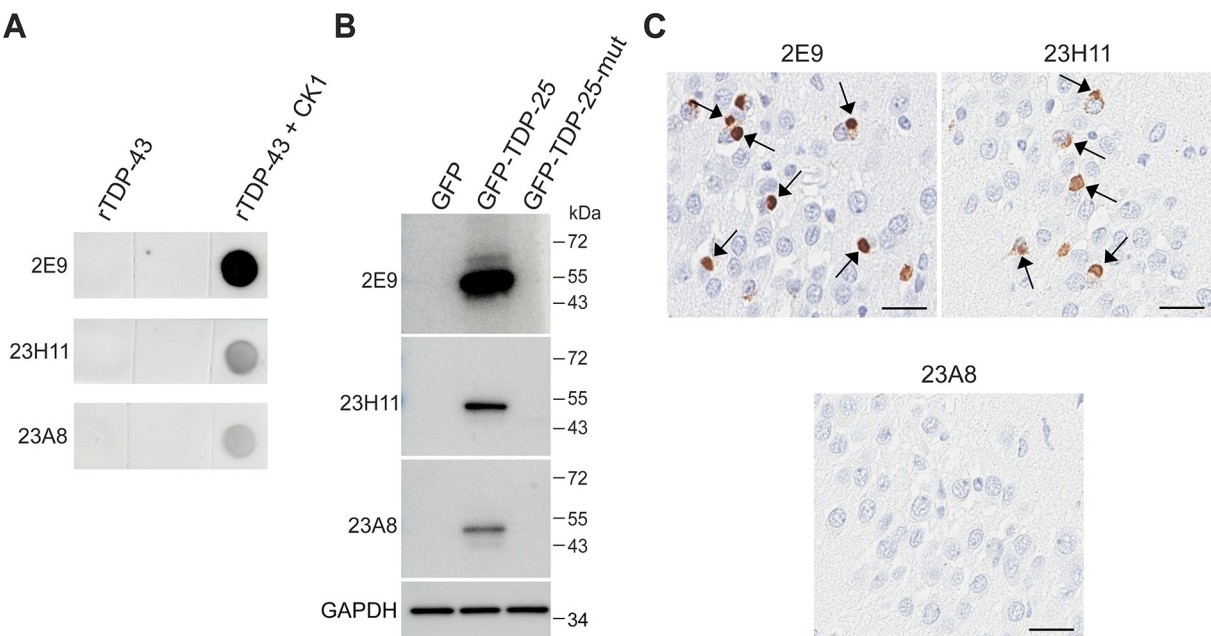

**Fig 2. Validation of representative B cell clones confirms their specificity for pS409/410-TDP-43.** (A) Dot blot analysis of rTDP43 treated with or without CK1 using the indicated supernatants from B cell clones. (B) Immunoblot analysis of HEK293T cell lysates expressing GFP, GFP-TDP-25, or GFP-TDP-25mut (S409A/S410A) using the indicated supernatants from B cell clones. GAPDH was used as a loading control. (C) Representative images of immunohistochemical analysis using the supernatants from indicated B cell clones in the hippocampus of human FTLD patients. Arrows mark TDP-43 inclusions. Scale bars are 20 μm.

Primary ELISA screening showed that eighty-nine B cell clones were immunoreactive for pS409/410-TDP-43 peptides at OD450, using a cutoff of 0.3 (S1 Fig) (S2 Table). Counter screening showed that these B cell clones have a stronger immunoreactivity to pS409/410-TDP-43 peptides than singly phosphorylated pS409- or pS410-TDP-43 peptides (S2 Table). Notably, most of the B cell clones have only minimum immunoreactivity for non-phosphorylated S409/410-TDP-43 peptides (NP-TDP-43; S2 Table), indicating their specificity. To further characterize these B cell clones, we selected forty-five clones that had an OD450 cutoff of 1.0 in the primary ELISA screen and assessed them by dot blot (S1 Fig). We found that thirty-nine of the forty-five B cell clones specifically detected CK1-treated rTDP-43 proteins, but not untreated rTDP-43, confirming their specificity for phosphorylated TDP-43. Variations in the degree of immunoreactivity among different clones were noted. For instance, as shown in Fig 2A, while all three representative clones (2E9, 23H11, and 23A8) detected CK1-treated rTDP-43, 2E9 showed stronger immunoactivity compared to 23H11 and 23A8 (Fig 2A). Based on their immunoactivity to detect CK1-treated rTDP-43, we selected fourteen clones for further evaluation (S1 Fig). Immunoblot analysis showed that twelve of the fourteen clones specifically detected GFP-TDP-25, but not GFP alone or GFP-TDP-25mut, confirming their specificity for pS409/410-TDP-43. Consistent with the dot blot results, 2E9 showed stronger immunoactivity to GFP-TDP-25 than 23H11 and 23A8 (Fig 2B). Finally, immunohistochemistry analysis showed that thirteen of the fourteen clones detected TDP-43 pathology in FTLD-TDP patient brains (S1 Fig). As shown in Fig 2C, 2E9 and 23H11, but not 23A8, detected cytoplasmic TDP-43. Based on the results of our dot blot, immunoblot, and immunohistochemistry analyses, we selected three promising B cell clones (26H10, 2E9, and 23A1) and used them to generate rabbit mAbs (S1 Fig). Notably, our findings establish the screening

workflow we utilized as an efficient method for identifying B cell clones that produce antibodies with strong specificity and sensitivity against pS409/410-TDP-43.

## Generation and characterization of rabbit mAbs against pS409/410-TDP-43

To generate rabbit mAbs, cDNA fragments encoding individual heavy and light chains were amplified by RT-PCR and inserted into mammalian expression vectors. Corresponding vectors were co-transfected into Expi293F cells, and the culture media was collected for affinity purification of rabbit mAbs. ELISA immunoassays showed that all three mAbs have immunoreactivity for pS409/410-TDP-43 peptides, but not NP-TDP-43 peptides (Fig 3A), confirming their specificity. Interestingly, 26H10 and 23A1, but not 2E9, also have immunoreactivity for singly phosphorylated pS409-TDP-43 peptides (Fig 3A). However, none of these three mAbs exhibited immunoreactivity for pS410-TDP-43 peptides (Fig 3A). Dot blot and immunoblot analyses showed that all three mAbs specifically detect CK1-treated rTDP-43 and GFP-TDP-25 when expressed in HEK293T cells (Fig 3B and 3C). In contrast, the three mAbs did not detect untreated rTDP-43 or GFP-TDP-25$_{mut}$ (Fig 3B and 3C), confirming their specificity

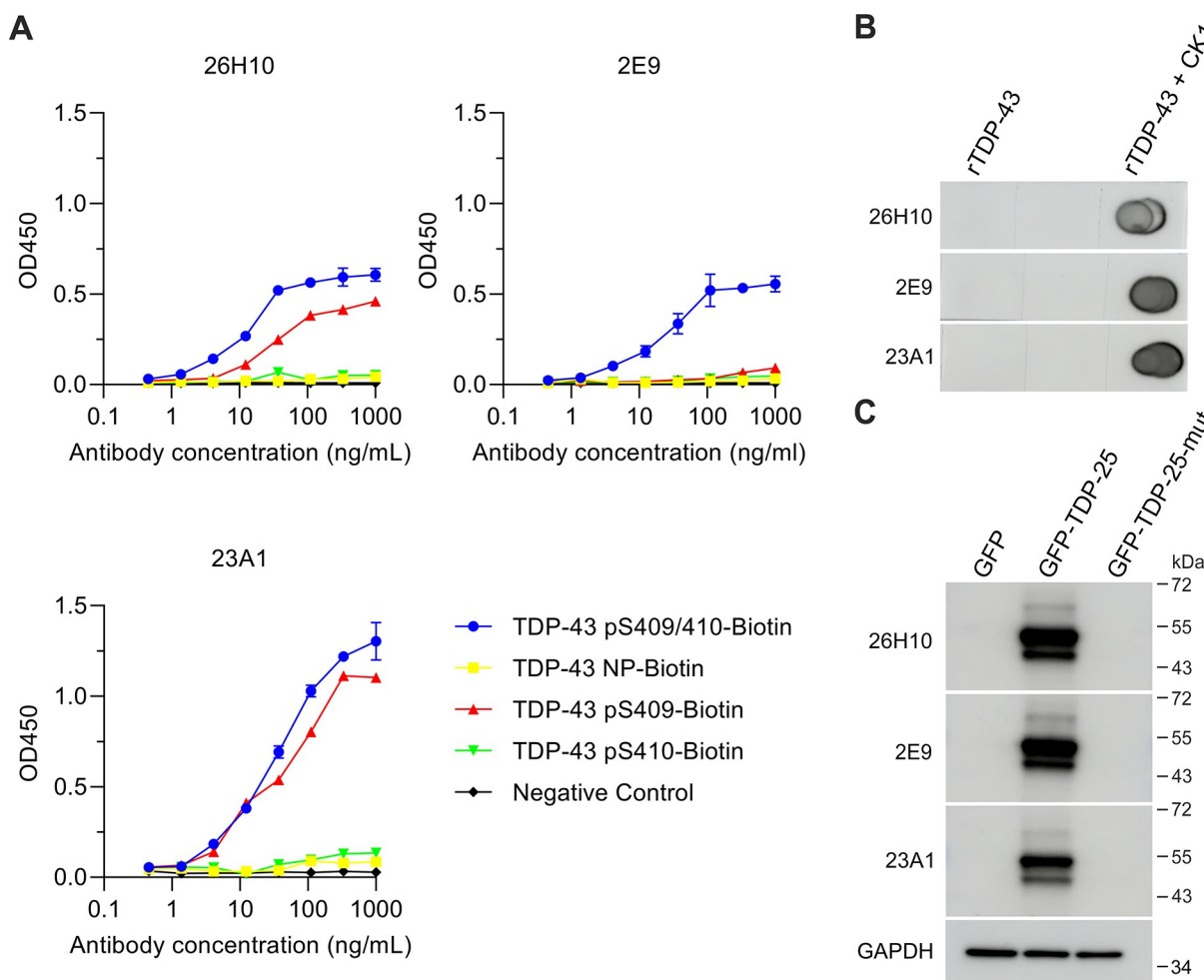

**Fig 3. Three rabbit mAbs exhibit high specificity and sensitivity to pS409/410-TDP-43.** (A) ELISA analysis of phospho- and nonphospho-TDP-43 peptides using the indicated rabbit mAbs. (B) Dot blot analysis of rTDP43 treated with or without CK1 using the indicated rabbit mAbs. (C) Immunoblot analysis of HEK293T cell lysates expressing GFP, GFP-TDP-25 or GFP-TDP-25$_{mut}$ (S409/410A) using the indicated rabbit mAbs. GAPDH was used as a loading control.

against pS409/410-TDP-43. To further confirm the specificity of our mAbs, we generated a truncated version of TDP-43 that lacks both amino acids 409 and 410 and the four residues beyond. In addition to wild-type GFP-TDP-43$_{1-408}$, we also generated a GFP-TDP-43$_{1-408-NLSmut}$ plasmid that harbors a mutated nuclear localization signal (NLS). This mutation causes TDP-43 to accumulate in the cytoplasm, mimicking disease-associated changes in TDP-43 localization and phosphorylation [21,22]. Immunoblot analysis of HEK293T cell lysates showed that our 26H10 mAb detects the phosphorylation of wild-type TDP-43, and this phosphorylation was more robust in the cells expressing GFP-TDP-43$_{NLSmut}$ (S2 Fig). In contrast, the 26H10 antibody did not detect GFP-TDP-43$_{1-408}$ or GFP-TDP-43$_{1-408-NLSmut}$ (S2 Fig), further confirming its specificity for pS409/410-TDP-43.

Given that accumulation of pS409/410-TDP-43 in the urea-soluble fraction of tissue lysates is a biochemical feature of FTLD-TDP [7,9,15], we evaluated whether our three mAbs could detect pathological TDP-43 in the urea-soluble fraction of FTLD-TDP brains. Immunoblots showed that, while an antibody against total TDP-43 (tTDP-43) detected comparable levels of TDP-43 in normal controls and FTLD-TDP patients (S3A Fig), all three mAbs detect pS409/410-TDP-43, including both full length and C-terminal fragments, in FTLD-TDP patients, but not in normal controls (Fig 4A). MSD immunoassays confirmed that all three mAbs detect an accumulation of pS409/410-TDP-43 in FTLD-TDP patients, as the pS409/410-TDP-43 signal was significantly higher in patient samples compared to normal controls (Fig 4B). Further evaluation in additional FTLD-TDP and ALS patient samples revealed that our three mAbs can detect different morphologic subtypes of TDP-43 pathology including neuronal cytoplasmic inclusions (NCI), neuronal intranuclear inclusions (NCII), dystrophic neurites (DN), and diffuse granular neuronal cytoplasmic inclusions (dNCI); staining was absent in normal controls (Figs 5A–5C and S3B). Finally, we evaluated whether our three mAbs could detect phosphorylated TDP-43 in mouse models of TDP-43 proteinopathy. We began with the rNLS8 transgenic model, which expresses human TDP-43 that lacks a NLS (hTDP-43ΔNLS). Consistent with previous reports showing that rNLS8 mice develop phosphorylated TDP-43 pathology [22,23], numerous cytoplasmic inclusions were detected in the cortex of 9-week-old rNLS8 mice using our three mAbs (Fig 5D). We then evaluated whether our three mAbs could also detect the phosphorylation of endogenous Tdp-43 in mice using the AAV-149R mouse model, which expresses a pathological 149 *C9orf72*-G$_4$C$_2$ repeat expansion and develops endogenous phosphorylated Tdp-43 pathology [24]. This pathology is absent in AAV-2R controls [24]. We found that 26H10 has the strongest immunoactivity against endogenous mouse pS409/410-Tdp-43 in AAV-149R mice and showed no immunoreactivity against Tdp-43 in AAV-2R controls (Fig 5E). In contrast, 2E9 detected considerably less endogenous phosphorylated Tdp-43 inclusions in the AAV-149R mice (Fig 5E), and 23A1 failed to detect any (Fig 5E). Taken together, these findings indicate that we successfully generated three rabbit mAbs that specifically detect pS409/410-TDP-43 in multiple model systems.

## Discussion

TDP-43 has been identified as the major component of the inclusions observed in approximately 50% of FTD patients and most ALS cases [1,2]. One of the disease-specific biochemical signatures of TDP-43 inclusions is abnormal phosphorylation [1,2]. Several aberrantly phosphorylated sites have been identified in FTD/ALS patients through the generation of phosphorylation site-specific antibodies and proteomic studies [9,25,26]. Mounting evidence supports the hypothesis that phosphorylation of TDP-43 at S409/410 residue sites is the most important disease hallmark in FTD/ALS patients and mouse models [7,9,15,22,23]. In this study, we generated three rabbit mAbs against pS409/410-TDP-43 and comprehensively

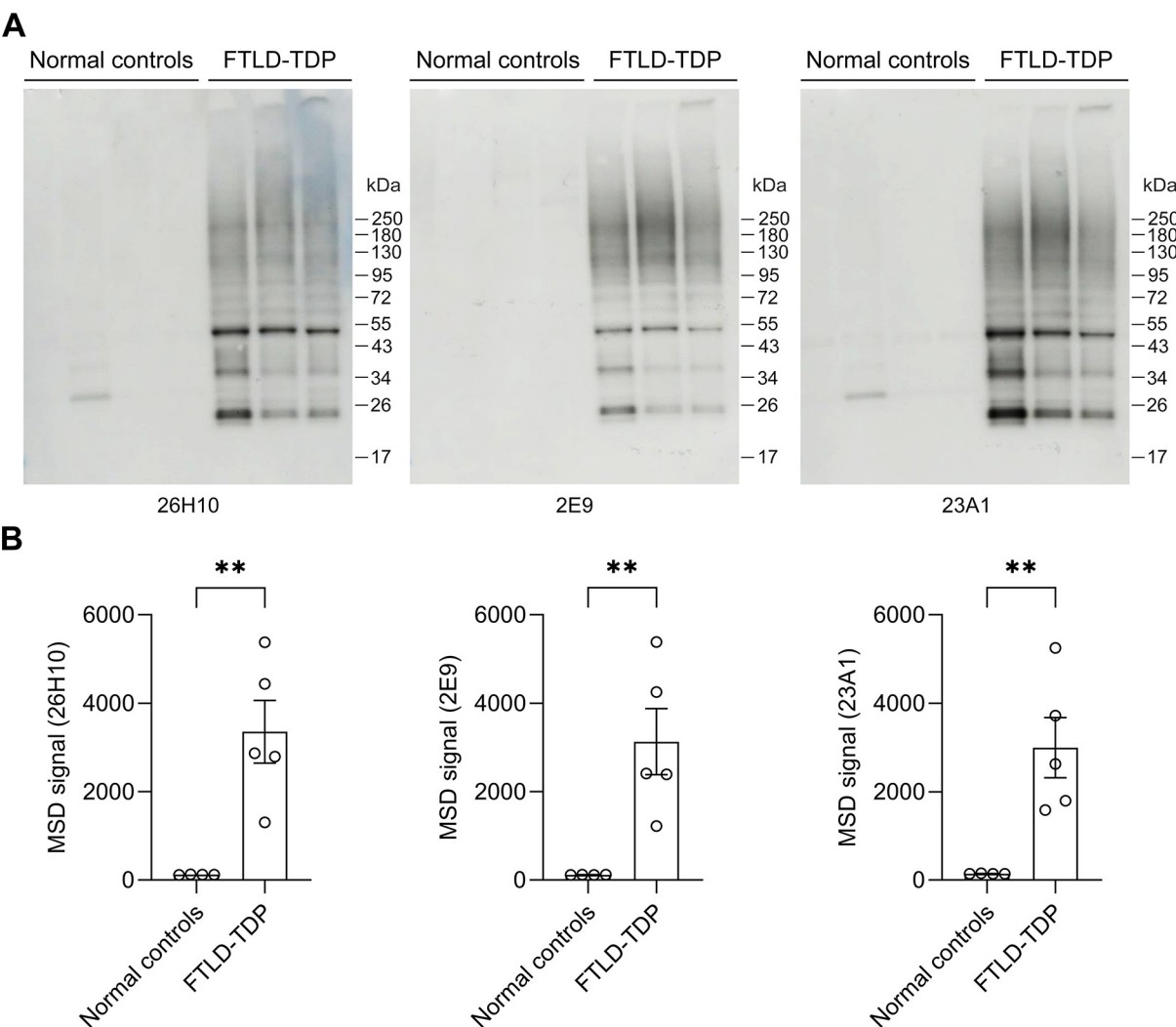

**Fig 4. Rabbit mAbs against pS409/410-TDP-43 detect the pathological accumulation of phosphorylated TDP-43 in FTLD-TDP brain tissues.** (A) Immunoblot analysis of urea-soluble fractions from the frontal cortex of FTLD-TDP patients and normal controls using the indicated rabbit mAbs. (B) MSD analysis of phosphorylated TDP-43 protein levels in urea-soluble fractions from the frontal cortex of FTLD-TDP patients and normal controls using the indicated rabbit mAbs (n = 4–5 per group). Data shown as the mean ± SEM. ** (left to right) P = 0.0050, 0.0091 and 0.0075, unpaired two-tailed t-test.

evaluated these mAbs in multiple model systems and in FTD/ALS patient samples. Our findings indicate that all three mAbs exhibit high specificity and sensitivity against pS409/410-TDP-43 and are ideal for use in multiple assays including biochemistry, immunohistochemistry, and immunoassays. Our findings also indicate that all there mAbs can be applied to multiple model systems including recombinant proteins, cultured cells, human samples, and mouse tissues. Moreover, compared to commercially available rabbit pS409/410-TDP-43 antibodies, our antibodies are monoclonal, and were generated and purified from the culture media of Expi293F cells transfected with specific heavy and light chain plasmids. Therefore, our mAbs are expected to show high reproducibility with minimal batch effects.

While generating our antibodies, we established a robust workflow to comprehensively screen and characterize B cell clones. The first step in this screening process uses ELISA immunoassays, which we have found to be a powerful high-throughput approach capable of sorting

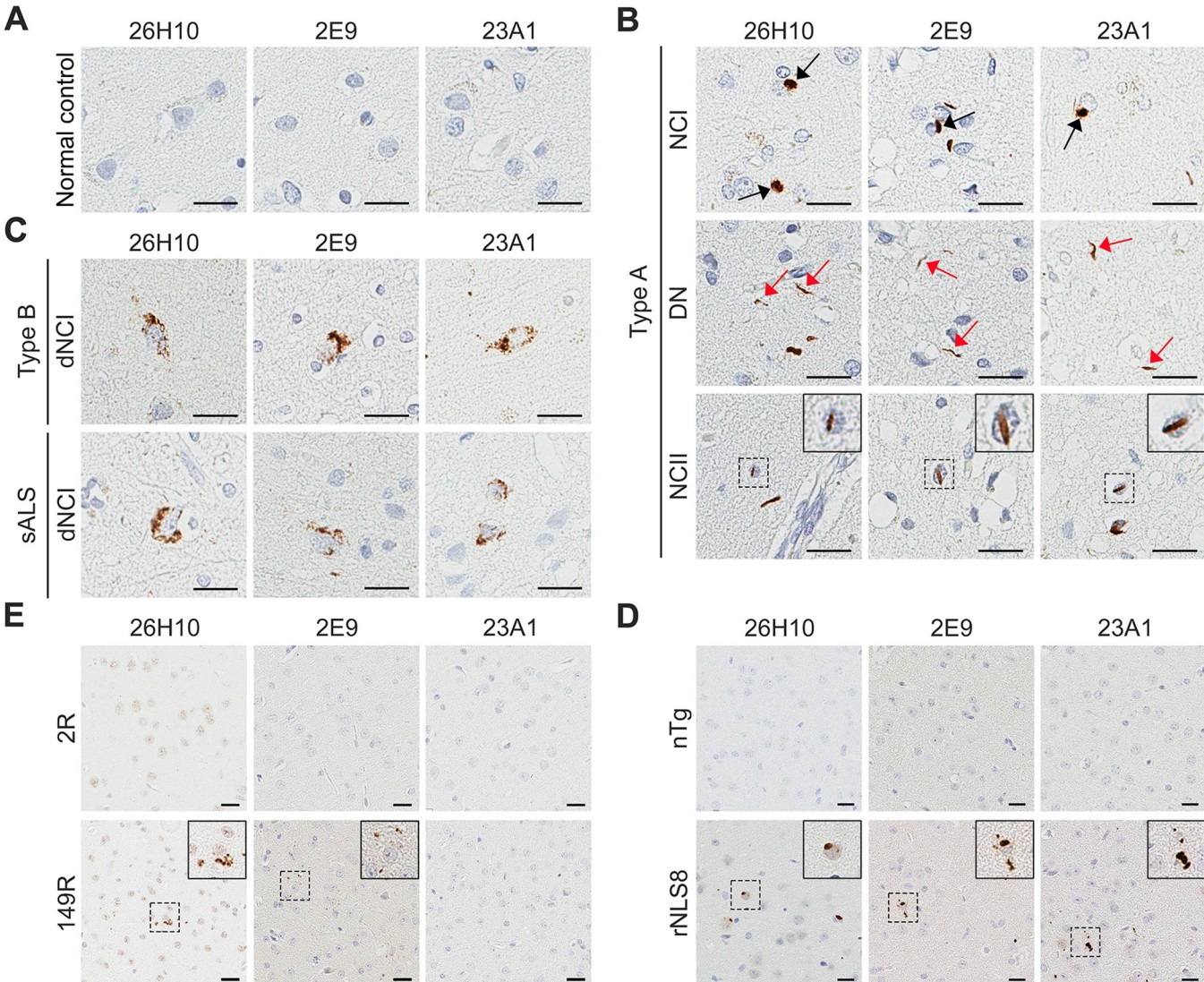

**Fig 5. Rabbit mAbs against pS409/410-TDP-43 detect TDP-43 pathology in brain tissues from FTLD and ALS patients and rNLS8 mice.** (A–C) Representative images of immunohistochemical analysis using the indicated rabbit mAbs against pS409/410-TDP-43 in the frontal cortex of normal controls (A), FTLD-TDP type A patients (B), and FTLD-TDP type B patients (C), and in the motor cortex of ALS patients (C). Black arrows indicate neuronal cytoplasmic inclusions (NCI), and red arrows mark dystrophic neurites (DN). Inserts in B are higher magnifications of neuronal intranuclear inclusions (NCII). (D) Representative images of immunohistochemical analysis using the indicated rabbit mAbs against pS409/410-TDP-43 in the cortex of non-transgenic (nTg) and rNLS8 mice. Inserts are higher magnifications of NCI. (E) Representative images of immunohistochemical analysis using the indicated rabbit mAbs against pS409/410-TDP-43 in the cortex of AAV-2R and AAV-149R mice. Inserts are higher magnifications of inclusions. For all panels, scale bars are 20 μm.

large numbers of samples (e.g., the 2688 B cell clones). Furthermore, the ELISA results are supported by other approaches. For instance, clone 2E9 showed a high immunoreactivity by ELISA and by dot blot (Fig 2), while clones 23A8 and 23H11 showed low immunoreactivity by ELISA and by dot blot (Fig 2). After primary screening by ELISA, secondary screening by biochemical assays like dot blots and immunoblots is necessary to eliminate false positives. In our hands, we found that six out of the forty-five selected clones failed to detect either phosphorylated or non-phosphorylated rTDP-43 proteins via dot blot, indicating a false positive rate of approximately 13.3%. As the third step in our workflow, we used a phosphorylation-resistant

construct to validate antibody specificity. For instance, we found two clones that specifically detected CK1-treated rTDP-43 (and failed to detect non-phosphorylated rTDP-43) in our dot bot analyses but exhibited weak immunoactivity to GFP-TDP-25$_{mut}$. As the latter is not phosphorylated at serines 409/410, these two clones are presumably reactive to other phosphorylated sites in TDP-43. Lastly, given that TDP-43 inclusions are the pathological hallmarks of FTD/ALS [1,2,7,9,15], it is important to evaluate whether individual B cell clones produce antibodies for immunohistochemistry applications. Indeed, we found that thirteen of the fourteen clones we evaluated specifically detected TDP-43 inclusions in FTD/ALS patient tissue (approximately a 92.9% positive rate).

After screening several B cell clones, we selected three clones for antibody generation and used the same approaches employed in our screen to comprehensively characterize three affinity-purified mAbs. Our results indicate that all three antibodies have high specificity and sensitivity against pS409/410-TDP-43. Interestingly, 26H10 and 23A1 also have immunoreactivity for singly phosphorylated pS409-TDP-43 peptides in ELISA assays, although this activity is less robust than their immunoreactivity for pS409/410-TDP-43 (Fig 3). In contrast, 2E9 does not have immunoreactivity for pS409-TDP-43 peptides (Fig 3), suggesting that our three antibodies have different secondary conformations. This hypothesis is further supported by the observation that our three antibodies have different sensitivities to endogenous phosphorylated mouse Tdp-43. Finally, we have further characterized our 26H10 antibody to confirm its specificity using truncated plasmids, and we assessed its sensitivity across different working dilutions in immunohistochemistry analyses. Importantly, immunotherapy using mAbs to target misfolded proteins has been widely investigated to treat neurodegenerative diseases. In fact, it has recently been reported that immunization with pS409/410-TDP-43 peptides, as a passive immunotherapy approach, reduces neuroaxonal damage in a TDP-43 mouse model [27]. We therefore believe our antibodies have therapeutic potential for the treatment of FTD/ALS and other TDP-43 proteinopathies. We already have the cDNA sequence of the heavy and light chains of each antibody, and we will focus on generating fragment antigen-binding region (Fab) fragments from our three mAbs in the future. We aim to test the therapeutic potential of these antibody fragments in TDP-43 mouse models.

## Conclusions

The three novel mAbs we developed are valuable tools for the research and diagnostic evaluation of TDP-43 pathology, and they have potential as immunotherapy agents for treating FTD/ALS.

## Supporting information

**S1 Fig. Schematic representation of the methods and procedures to generate, screen and characterize monoclonal antibodies against pS409/410-TDP-43.**
(DOCX)

**S2 Fig. 26H10 rabbit mAb exhibits high specificity to pS409/410-TDP-43.**
(DOCX)

**S3 Fig. 26H10 rabbit mAb detects TDP-43 pathology in FTLD-TDP brain tissues.**
(DOCX)

**S1 Table. Characteristics of patients with FTD/ALS.**
(DOCX)

**S2 Table. Immunoreactivity of B cell clones against TDP-43 species as measured by ELISA immunoassay.**
(DOCX)

**S1 Raw images.**
(PDF)

## Acknowledgments

We are grateful to all patients and their families who agreed to donate post-mortem tissue.

## Author Contributions

**Conceptualization:** Leonard Petrucelli, Yong-Jie Zhang.

**Funding acquisition:** Tania F. Gendron, Dennis W. Dickson, Leonard Petrucelli, Yong-Jie Zhang.

**Investigation:** Paula Castellanos Otero, Wei Shao, Caroline J. Jones, Kexin Huang, Lillian M. Daughrity, Mei Yue.

**Resources:** Udit Sheth, Tania F. Gendron, Mercedes Prudencio, Björn Oskarsson, Dennis W. Dickson.

**Supervision:** Leonard Petrucelli, Yong-Jie Zhang.

**Writing – original draft:** Paula Castellanos Otero, Tiffany W. Todd, Leonard Petrucelli, Yong-Jie Zhang.

**Writing – review & editing:** Tiffany W. Todd, Leonard Petrucelli, Yong-Jie Zhang.

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
