## [Decision Letter · Decision Letter 0]

1 Dec 2023

PONE-D-23-36865Generation and characterization of monoclonal antibodies against pathologically phosphorylated TDP-43PLOS ONE

Dear Dr. Zhang,

Thank you for submitting your manuscript to PLOS ONE, and personal apologies for the delay in getting back to you.Despite the requirement of two reviewers, I will go ahead with only one report since its constructive opinion summarizes all items that must be either amend or explain.After careful consideration, we feel that it has merit but does not fully meet PLOS ONE’s publication criteria as it currently stands. Therefore, we invite you to submit a revised version of the manuscript that addresses the points raised during the review process.

We look forward to receiving your revised manuscript.

Kind regards,

Maria Gasset, Ph.D.

Academic Editor

PLOS ONE

Journal Requirements:

2. "We note that the grant information you provided in the ‘Funding Information’ and ‘Financial Disclosure’ sections do not match. 

Reviewers' comments:

Reviewer's Responses to Questions

**Comments to the Author**

1. Is the manuscript technically sound, and do the data support the conclusions?

Reviewer #1: Yes

2. Has the statistical analysis been performed appropriately and rigorously? 

Reviewer #1: Yes

3. Have the authors made all data underlying the findings in their manuscript fully available?

Reviewer #1: Yes

4. Is the manuscript presented in an intelligible fashion and written in standard English?

Reviewer #1: Yes

5. Review Comments to the Author

Reviewer #1: In this brief manuscript, Zhang et al. describe the generation of novel antibodies directed against phospho-TDP-43 and provide compelling evidence of the antibodies´ validity detecting TDP-43 inclusions in situ. The data is clearly presented and the conclusions appear robust. The material presented here could be of interest and use in the field and the manuscript should be of interest to the reader. However, the manuscript could be further elaborated to provide a clear discussion positioning the novel antibodies with respect to the available ones and additional experiments could help to detail which species are really being detected. What follows is a list of issues which, in this reviewer´s opinion, could help improve the manuscript and categorize the described antibodies.

Major points:

#1. Please provide a clear description of the advantages of these novel antibodies with respect to the commercial ones, in particular in light of recent publications (Riemenschneider et al. https://doi.org/10.1186/s40478-023-01592-z).

#2. The major confusion raised while following the manuscript was trying to understand which species are specifically targeted by the antibodies. Although still open to debate, it could be concluded that phosphorylation of TDP-43 low complexity domain promotes insolubility (see da Silva et al. doi: 10.15252/embj.2021108443). In addition, CK1 phosphorylation may not be complete, which means that the antigens for immunization could correspond to aggregates containing mixtures of phosphorylated and non-phosphorylated TDP-43. In Figure 2, for instance, it is not clear if cell B supernatant is targeting aggregated (urea-soluble, in a broad way) rather than selectively phosphorylated TDP-43. In addition, Figure 4 suggests that the novel antibodies are targeting aggregated TDP-43. To clarify this, authors should demonstrate that aggregated, non-phosphorylated TDP-43 is not detected by the novel antibodies.

#3. Authors generate a double Ala mutant in positions 409/410 as a control of phosphorylation of these specific sites. Instead, authors should test the possible detection by the antibodies of a truncated version of TDP-43, where both 409 and 410 (and beyond) sites are not included.

Minor points:

#1. Please provide an explanatory scheme of the procedure followed to produce the antibodies and the specific epitopes which are targeted.

#2. Please include the sequence of the biotinylated peptides used in the dot-blots in Fig.2.

#3. Please include a loading control in the western blots (Calnexin, b-actin).

#4. Antibodies are used in 1:500 dilution in immunochemistry experiments. This appears too high. Please provide an explanation.

#5. Please show the uncropped gel in Fig. 1A, or at least the region covering molecular weights below 34 kDa. This is particularly relevant in Fig. 1B since the data with the doble Ala mutant appears crucial.

#6. In lines 219-220, reference to Fig 1B should instead be Fig 1C.

6. PLOS authors have the option to publish the peer review history of their article (what does this mean?). If published, this will include your full peer review and any attached files.

Reviewer #1: No

---

## [Author Response · Author response to Decision Letter 0]

16 Jan 2024

We included a point-by-point 'Response to Reviewers' to address the reviewer's concerns in our resubmission.

---

## [Editor Report · Decision Letter 1]

18 Jan 2024

Generation and characterization of monoclonal antibodies against pathologically phosphorylated TDP-43

PONE-D-23-36865R1

Dear Dr. Yong-Jie Zhang,

We’re pleased to inform you that your manuscript has been judged scientifically suitable for publication and will be formally accepted for publication once it meets all outstanding technical requirements.

Kind regards,

Maria Gasset, Ph.D.

Academic Editor

PLOS ONE
---

## [Editor Report · Acceptance letter]

27 Mar 2024

PONE-D-23-36865R1 

PLOS ONE

Dear Dr. Zhang, 

I'm pleased to inform you that your manuscript has been deemed suitable for publication in PLOS ONE. Congratulations! Your manuscript is now being handed over to our production team.

Kind regards, 

on behalf of

Dr. Maria Gasset 

Academic Editor

PLOS ONE